# Single and Mixed Strains of Probiotics Reduced Hepatic Fat Accumulation and Inflammation and Altered Gut Microbiome in a Nonalcoholic Steatohepatitis Rat Model

**DOI:** 10.3390/biomedicines12081847

**Published:** 2024-08-14

**Authors:** Maneerat Chayanupatkul, Panrawee Machchimapiro, Natthaya Chuaypen, Natcha Wanpiyarat, Somying Tumwasorn, Prasong Siriviriyakul, Duangporn Werawatganon

**Affiliations:** 1Center of Excellence in Alternative and Complementary Medicine for Gastrointestinal and Liver Diseases, Department of Physiology, Faculty of Medicine, Chulalongkorn University, Bangkok 10330, Thailand; maneeratc@gmail.com (M.C.); panrawee.ma@gmail.com (P.M.); fmedpsr@gmail.com (P.S.); 2Center of Excellence in Hepatitis and Liver Cancer, Department of Biochemistry, Faculty of Medicine, Chulalongkorn University, Bangkok 10330, Thailand; natthaya.c@chula.ac.th; 3Metabolic Diseases in Gut and Urinary System Research Unit (MeDGURU), Department of Biochemistry, Faculty of Medicine, Chulalongkorn University, Bangkok 10330, Thailand; 4Department of Pathology, Faculty of Medicine, Chulalongkorn University, Bangkok 10330, Thailand; natcha.w@chula.ac.th; 5Department of Microbiology, Faculty of Medicine, Chulalongkorn University, Bangkok 10330, Thailand; somying.t@chula.ac.th

**Keywords:** probiotics, nonalcoholic steatohepatitis, gut microbiota, TLR4, CD14

## Abstract

As gut dysbiosis has been implicated in the pathogenesis of nonalcoholic steatohepatitis (NASH), probiotic supplementation might be a potential treatment for this condition. The aim of this study was to evaluate the effects of single- and mixed-strain probiotics on the severity of NASH induced by a high-fat, high-fructose (HFHF) diet and their mechanisms of action. Male Sprague–Dawley rats were divided into four groups (*n* = 7 per group): control group, NASH group, NASH + single-strain group, and NASH + mixed-strain group. In the single-strain and mixed-strain groups, rats received *Lactobacillus plantarum* B7 and *Lactobacillus rhamnosus* L34 + *Lactobacillus paracasei* B13 by oral gavage once daily, respectively. The duration of the study was 6 weeks. Liver tissue was used for histopathology, hepatic fat content was assessed by Oil Red O staining and hepatic free fatty acid (FFA), and hepatic TLR4 and CD14 expression were assessed by immunohistochemistry. Fresh feces was collected for gut microbiota analysis. Liver histology revealed a higher degree of fat accumulation, hepatocyte ballooning, and lobular inflammation in the NASH group, which improved in probiotics-treated groups. The amounts of hepatic fat droplets and hepatic FFA levels were more pronounced in the NASH group than in the control and treatment groups. Serum TNF- α levels were significantly higher in the NASH group than in control and probiotic groups. The expression of CD14 and TLR4 increased in the NASH group as compared with the control and probiotics-treated groups. Alpha diversity was reduced in the NASH group, but increased in both treatment groups. The relative abundance of *Lactobacillus* significantly decreased in the NASH group, but increased in both treatment groups. The relative abundance of *Akkermansia* significantly increased in the NASH group, but decreased in both treatment groups. In conclusion, both single-strain and mixed-strain probiotics could improve NASH histology by suppressing inflammatory responses in the liver, with this improvement potentially being associated with changes in the gut microbiota.

## 1. Introduction

Nonalcoholic fatty liver disease (NAFLD) is a hepatic manifestation of metabolic syndrome. It includes a spectrum of pathological changes such as simple steatosis, nonalcoholic steatohepatitis (NASH), fibrosis, cirrhosis, and in some cases, hepatocellular carcinoma [1]. High fat and high fructose (HFHF) consumption, a sedentary lifestyle, and environmental and genetic factors lead to the development of insulin resistance and metabolic syndrome. Insulin resistance and adipose tissue dysfunction subsequently result in hepatic fat accumulation, mitochondrial dysfunction, oxidative stress, and endoplasmic reticulum stress. All these changes are fundamental to the development of NASH [1]. Recent studies suggest that the alteration of gut microbiota and increased intestinal permeability that occur because of obesogenic food intake also play an important role in the pathogenesis of NAFLD [2,3]. The development of bacterial translocation and endotoxemia from an impaired intestinal barrier lead to hepatic inflammation and injury, likely through the activation of toll-like receptor-4 (TLR4) [4].

The binding of lipopolysaccharide (LPS) and other endogenous ligands, such as free fatty acids (FFAs), to TLR4 results in nuclear translocation of nuclear factor-κB (NF-κB) and stimulates downstream inflammatory cascades, leading to the release of proinflammatory cytokines, such as tumor necrosis factor-alpha (TNF-α) and interleukin-6 (IL-6). TLR4 expression is also up-regulated in the presence of LPS and FFAs [5]. Moreover, TLR4 activation might have a direct role in hepatic fat accumulation, as evidenced by the increased fatty acid oxidation and reduced liver triglyceride levels in TLR4-deficient mice [6]. CD14 is a pattern recognition molecule found in innate immune cells, such as monocytes, that recognizes LPS [7]. Upon binding with its LPS, CD14 transports LPS to TLR4 and promotes the endocytosis of TLR4, which is a crucial step for the stimulation of its downstream signaling [8]. An animal study showed that hepatic stellate cells (HSCs) isolated from leptin-deficient and hyperleptinemic mice had higher expression of CD14, more pronounced response to LPS, and stronger inflammatory and fibrogenic reactions than lean mice [9]. Moreover, a human study also showed that serum-soluble CD14 levels were significantly correlated with liver inflammation and fibrosis in patients with NAFLD [10].

Given the role of gut dysbiosis in the pathogenesis of NASH, probiotic supplements have been studied in both preclinical and clinical studies of NASH to evaluate their effects on liver histology, inflammatory markers, oxidative stress markers, lipid synthetic gene expression, and gut microbiota. Commonly used probiotics in those studies are *Bifidobacterium* and *Lactobacillus,* either as single or multiple strains. In rodent studies, probiotics generally alleviate the severity of liver histology, but their effects on liver enzymes and gut microbial composition are mixed [11]. The effects of probiotics on NAFLD in clinical studies have been encouraging, but not as robust as animal studies, particularly in terms of liver histology [12]. Moreover, there is significant heterogenicity in terms of the strains and doses of probiotics used in each study. In this study, we aimed to determine in the effects of *Lactobacillus plantarum* B7 (*L. plantarum* B7) and the combination of *Lactobacillus rhamnosus* L34 (*L. rhamnosus* L34) and *Lactobacillus paracasei* B13 (*L. paracasei* B13) on liver histology, hepatic fat content, hepatic expression of TLR4 and CD14, serum TNF-α levels, and fecal microbiota in a rat model of NASH.

*L. plantarum* EMCC-1039 [13] and *L. plantarum* ATG-K2/6 [14] have been shown to reduce hepatic steatosis, serum ALT, and hepatic triglyceride levels. A previous study demonstrated that *L. plantarum* B7 could reduce hepatic steatosis and liver inflammation through the increased expression of peroxisome proliferator-activated receptors gamma (PPAR-γ) and reduced oxidative stress [15]. Moreover, *L. plantarum* B7 has been shown to improve liver histopathology, reduce hepatic TLR4 expression, and ameliorate gut dysbiosis in a rat model of alcohol-induced liver injury [16]. Therefore, in this study, we aimed to explore the effects of *L. plantarum* B7 on gut microbiome-related mechanisms and their implications for NASH development. *L. rhamnosus* and *L. paracasei,* either alone or in combination, have been shown in previous animal studies to improve hepatic steatosis; however, their effects on serum ALT and AST have been mixed [11]. Our strains of interest (*L. rhamnosus* L34 and *L. paracasei* B13) have never been studied in NAFLD. Nevertheless, an in vitro study demonstrated that *L. rhamnosus* L34 and *L. paracasei* B13 (previously identified as *L. casei*) could reduce *Clostridioides difficile*-induced IL-18 production in HT-29 cells through the suppression of phospho-NF-κB and phospho-c-JUN activation [17]. Moreover, *L. paracasei* B13 (previously identified as *L. casei*) has previously been shown to reduce bacterial translocation, serum IL-6, and mortality in a murine sepsis model [18]. Our previous research in a rat model of alcohol-induced liver injury also showed that the combination of *L. rhamnosus* L34 and *L. paracasei* B13 (previously identified as *L. casei* L39) could improve liver histology, reduce serum TNF-α levels, suppress hepatic TLR4 expression, and promote the growth of beneficial bacteria [16]. Based on the results of previous studies, we hypothesized that the combination of *L. rhamnosus* L34 and *L. paracasei* B13 might improve gut dysbiosis, reduce liver inflammation, and thus improve NASH histopathology.

## 2. Materials and Methods

### 2.1. Animals and Experimental Design

Male Sprague–Dawley rats at 6 weeks of age (180–220 g) were purchased from the Nomura Siam International Co., Ltd., Bangkok, Thailand and housed under standard conditions (12 h light/dark cycle and room temperature of 25 ± 1 °C). After 1 week of acclimatization, rats were divided into four groups (*n* = 7 per group). In the control group, rats were fed ad libitum with a standard rat chow diet; in the NASH group, rats were fed ad libitum with an HFHF diet containing 55% fat, 10% protein, and 35% carbohydrate (20% fructose and 15% starch) for 6 weeks; in the NASH + single strain group, rats were fed ad libitum with an HFHF diet for 6 weeks and simultaneously received 1 mL of *L. plantarum* B7 suspension (1.8 × 10^9^ CFU/mL) by oral gavage once daily during these 6 weeks; and in the NASH + mixed strains group, rats were fed ad libitum with an HFHF diet for 6 weeks and simultaneously received 1 mL of the combination suspension, which included *L. rhamnosus* L34 (1.8 × 10^9^ CFU/mL) + *L. paracasei* B13 (1.8 × 10^9^ CFU/mL), by oral gavage once daily during these 6 weeks (with a total bacteria dosage of 1.8 × 10^9^ CFU/mL). We did not combine *L. plantarum* B7 with *L. rhamnosus* L34 and *L. paracasei* B13 because, according to our unpublished data, *L. plantarum* B7 suppressed the growth of other bacteria when used in combination. At the end of the experiment, all rats were euthanized with an intraperitoneal injection of overdose sodium pentobarbital. All animal care was conducted in accordance with the Ethical Principles and Guidelines for the Use of Animals by the National Research Council of Thailand, and the experimental protocol was approved by the Animal Care and Use Committee, Faculty of Medicine, Chulalongkorn University, Bangkok, Thailand (Approval number: 015/2563). This study was reported in accordance with Animal Research: Reporting of In Vivo Experiments (ARRIVE) guidelines. Sample size was calculated using data from a study by Werawatganon and colleagues [15] Appendix A. The formula for the HFHF diet was adopted from a study by Witayavanitkul and colleagues [19].

### 2.2. Probiotics Preparation

*L. plantarum* B7, *L. rhamnosus* L34, and *L. paracasei* B13 were isolated from infant feces [17] and stored in de Man–Rogosa–Sharp (MRS) broth (Oxoid, Basingstoke, United Kingdom) with 20% glycerol at −80 °C. These strains were recovered from frozen stock and cultivated twice on MRS agar (10% CO_2_, 10% H_2_, and 80% N_2_) at 37 °C in an anaerobic jar for 48 h. A single colony of these bacteria was sub-cultured on MRS broth and grown at 37 °C under anaerobic conditions for 48 h. The plate was in an Anaerobe box (Thermo Scientific, USA) with an Anaero pack (Thermo Scientific, Waltham, MA, USA) for 48 h. The products were reconstituted in 0.9% sterile water and adjusted to the concentration to 1.8 × 10^9^ CFUs/mL by a spectrophotometer (Bio-Rad Smart SpecTM Plus, Hercules, CA, USA).

### 2.3. Liver Histopathological Evaluation

The right lobe of the liver was used for the histological evaluation. Liver tissue samples were fixed in 10% formalin for 24 h and placed in paraffin embedding cassettes. Fixed liver tissues were then cut into 4 mm thick paraffin sections and deparaffinized with xylene. Then, they were hydrated in 95% alcohol for 5 min, washed with water and stained with hematoxylin and eosin (H&E). Slides were visualized under a light microscope for the grading of hepatic steatosis, lobular inflammation, and hepatocyte ballooning according to Brunt’s criteria as described below [20].

Steatosis: Grade 0 = absence of fat, 1 = < 33% of hepatocytes with fat droplets, 2 = 33–66% of hepatocytes with fat droplets, 3 = > 66% of hepatocytes with fat droplets.

Inflammation: Grade 0 = no inflammation, 1 = < 2 foci per 20 × field, 2 = 2–4 foci per 20 × field, 3 = > 4 foci per 20 × field.

Hepatocyte ballooning: Grade; 0 = no ballooning, 1 = few ballooning cells, 2 = many ballooning cells.

An experienced pathologist who assessed and graded liver histology was blinded to the experimental groups. For steatosis grading, the pathologist examined the entirety of four different histology slides in each group and estimated the percentage of hepatocytes with intracellular fat droplets.

### 2.4. Hepatic Lipid Accumulation Evaluation

Liver was embedded in a compound with the proper temperature (OTC, Sakura Finetek, Torrance, CA, USA). Tissue was sectioned at 4 µm thickness using Microtome Cryostat. Liver tissues were stained with fresh 60% Oil Red O working solution O (Sigma-Aldrich, USA) for 7 min, followed by counter-staining with hematoxylin for 2 min. Total fat accumulation was examined under a bright-field microscope. The percentage of fat deposition was calculated using color histogram analysis by Image J software version 1.54a (NIH, New York, NY, USA).

### 2.5. Immunohistochemistry

Deparaffinization was performed by incubating slides with xylene 3 times for 10 min each time and rehydrating with ethanol in a graded series. For the antigen retrieval process, sections were cooked in a retrieval buffer using an automated retriever (Dako, Carpinteria, CA, USA). The endogenous peroxidase was blocked with 3% hydrogen peroxide in methanol (Merck, Darmstadt, Germany) for 5 min, and non-specific background staining was blocked with goat serum (Dako, Carpinteria, CA, USA) diluted in PBS for 20 min. Three sections were cut for immunohistochemistry (IHC). Slides were incubated with primary antibodies against TLR4 (Bio-Rad Laboratories, Hercules, CA, USA) and CD14 (Elabscience Biotechnology, Houston, TX, USA) with the concentration of 1:100 at 4 °C overnight. Subsequently, slides were incubated with secondary antibodies. All antibodies, reagents, and equipment, except for anti-TLR4 and anti-CD14 antibodies, were obtained from Dako, USA. After counterstaining with hematoxylin, slides were cover-slipped. Histologic slides were scanned at a 40× resolution using a whole-slide scanner with the Aperio ScanScope CS system, and images were analyzed using the Aperio ImageScope software version 12.3.3 (Leica Biosystems Imaging, San Diego, CA, USA). Positive cells were those with brown strained color in the cytoplasm. Percent positivity was calculated as the number of positive pixels divided by the total number of pixels × 100. Positive and negative pixels were set by using positive and negative control tissues [21].

### 2.6. Hepatic Free Fatty Acid Measurement

Fresh liver tissue was harvested, placed in liquid nitrogen, and stored at −80 °C until the time of analysis. Liver tissue was homogenized in 200 μL of chloroform/Triton X-100 and incubated on ice for 30 min. The extract was centrifuged for 10 min using a microcentrifuge. The organic phase was collected and air-dried at 50 °C, followed by vacuum drying to remove chloroform. Dried lipids were dissolved in 200 μL of fatty acid assay buffer (Abcam, Cambridge, UK) and vortexed for 5 min. Hepatic free fatty acid (FFA) levels were quantified using a commercially available colorimetric assay kit according to the manufacturer’s manual (Abcam, Cambridge, UK). Hepatic free fatty acid levels were expressed as nmol/mg of tissue.

### 2.7. Serum TNF-α and IL-6 Measurement

Cardiac puncture was performed to obtain blood samples. Whole blood was centrifuged at 2000× *g* for 20 min at 4 °C to obtain serum samples. Serum TNF-α and IL-6 levels were quantified using commercially available ELISA kits for TNF-α and IL-6 according to the manufacturer’s instructions (R&D Systems, Minneapolis, MN, USA). Serum TNF-α and IL-6 levels were expressed as pg/mL.

### 2.8. Fecal Microbiota

At the end of the experiment, fresh stool samples were collected from each rat and stored at −80 °C until analysis. Stool samples were processed and analyzed with the ZymoBIOMICS^®^ Service: Targeted Metagenomic Sequencing (Zymo Research, Irvine, CA, USA). DNA extraction was performed using a ZymoBIOMICS^®^-96 MagBead DNA Kit (Zymo Research, Irvine, CA, USA) according to the manufacturer’s instructions and amplified at the V3-V4 regions of 16srRNA gene using the polymerase chain reaction (PCR) method. DNA sequencing and taxonomy assignment were performed according to Zymo Research protocols.

### 2.9. Statistical Analysis

The Shapiro–Wilk test was used to test the normality of the continuous variables. The continuous data with normal distribution were presented as mean ± standard error of mean (SEM), whereas data with non-normal distribution were presented as median ± interquartile range (IQR). Continuous variables in each group were compared using either one-way analysis of variance (one-way ANOVA) with post hoc LSD or the Kruskal–Wallis test with post hoc Dunn’s test as appropriate using the Statistics Package for the Social Sciences (SPSS) software version 18.0 for Windows. Probability values of less than 0.05 were considered statistically significant.

Gut microbiota analysis: 16S rRNA gene-targeted sequencing with a V3-V4 hypervariable region was amplified and performed using the Quick-16S™ NGS Library Prep Kit (Zymo Research, Irvine, CA, USA). The library was sequenced on Illumina^®^ MiSeq™ with a v3 reagent kit (600 cycles). The sequencing was conducted with 10% PhiX spike-in. Unique amplicon sequences variants (ASVs) were inferred from raw reads using the DADA2 pipeline [22]. The DADA2 pipeline was performed to remove chimeric sequences. Taxonomy assignment was performed using Uclust from QIIME v2.0 with a reference database. Alpha diversity (Shannon diversity index) and composition visualization were calculated using Microbiome R packages (v1.20.0). Beta diversity was analyzed using principal coordinate analysis (PCoA) according to Bray–Curtis distances and permutational multivariate analysis of variance (PERMANOVA) [23]. DESeq2 was performed to identify the differential abundance of bacteria between groups [24]. To identify specific microbial communities that differed significantly among the groups, LEfSe (Linear discriminant analysis Effect Size) analysis was performed (linear discriminant analysis or LDA score > 3.0, *p*-value < 0.005 and False Discovery Rate (FDR)-adjusted *p*-value < 0.1). The correlation analyses were performed using R packages. P-values of less than 0.05 were considered statistically significant. PICRUSt2 (Phylogenetic Investigation of Communities by Reconstruction of Unobserved States) tool was used for functional profile prediction (https://github.com/picrust/picrust2, accessed on 4 August 2024).

## 3. Results

### 3.1. Effects of Single- and Mixed-Strain Probiotics on Liver Histology

In the NASH group, liver histology demonstrated a higher degree of liver steatosis (median score of 3 ± 0), ballooning (median score of 2 ± 0), and lobular inflammation (median score of 2 ± 0) than in the control group (median score of 0 ± 0 for all components) (Figure 1A, Table 1, *p* < 0.001 for all components). NASH scores were significantly higher in the NASH group than in the control group (7 ± 0 vs. 0 ± 0, respectively, *p* < 0.001, Figure 1B, Table 1). The degrees of liver steatosis (median score of 2 ± 1 in both probiotic groups) and inflammation (median score of 1 ± 1 in the single-strain group and 1 ± 0 in the mixed-strain group) were less severe (*p* < 0.05 for both components), but the severity of hepatocyte ballooning did not significantly improve in either probiotic group (median score of 2 ± 0 in the single-strain group and 2 ± 1 in the mixed-strain group) as compared with the NASH group. Administration of single and mixed strains of probiotics improved NASH scores as compared with the NASH group (5 ± 2 vs. 5 ± 2 vs. 7 ± 0, respectively, *p* < 0.05, Figure 1C,D, Table 1). Both treatments, however, did not normalize liver histology when compared to the control group, as some degree of liver steatosis, inflammation, and hepatocyte ballooning were still seen after treatment.

### 3.2. Effects of Single- and Mixed-Strain Probiotics on Hepatic Lipid Accumulation and Hepatic Free Fatty Acid Levels

Lipid droplets were stained red with Oil Red O staining and visualized by light microscopy at ×40 magnification, as shown in Figure 2A–D. Rats in the NASH group had a higher amount and larger size of lipid droplets compared with those in the control group. In both treatment groups, lipid droplets were smaller in size and less abundant than in the NASH group.

The red-stained pixels (representing fat droplet) were segmented using manual intensity thresholding. Figure 2E demonstrates the percentage (%) of fat, which was calculated as the ratio of red-stained pixels to the total number of pixels. A significant increase in an area with positive Oil Red O staining was observed in the NASH group compared with the control group (65.44 ± 1.31% vs. 1.41 ± 0.30%, respectively; *p* < 0.05). In the single- and mixed-strain groups, fat accumulation decreased as compared with the NASH group (10.49 ± 1.12% vs. 17.65 ± 1.76% vs. 65.44 ± 1.31%, respectively; *p* < 0.05). Hepatic fat accumulation was significantly more pronounced in the mixed-strain group than in the single-strain group (17.65 ± 1.76% vs. 10.49 ± 1.12%, respectively; *p* < 0.05). In addition, hepatic steatosis in both the single-strain and mixed-strain probiotic groups was higher than in the control group (10.49 ± 1.12% vs. 17.65 ± 1.76% vs. 1.41 ± 0.30%, respectively; *p* < 0.05).

As shown in Figure 2F, hepatic FFA levels were significantly higher in the NASH group than in the control group (13.92 ± 0.02 vs. 3.56 ± 0.22 nmol/mg, *p* < 0.001). Hepatic FFA levels were significantly lower in both probiotic groups when compared with the NASH group (6.51 ± 0.16 nmol/mg in the single-strain group and 8.01 ± 0.55 nmol/mg in the mixed-strain group, *p* < 0.001 for both comparisons). Hepatic FFA levels were still significantly higher in both probiotic groups than in the control group.

### 3.3. Effects of Single- and Mixed-Strain Probiotics on Serum TNF-α and IL-6 Levels

As shown in Figure 3A, serum TNF-α levels were significantly higher in the NASH group than in the control group (278.24 ± 4.89 vs. 44.20 ± 1.09 pg/mL, respectively, *p* < 0.001). In both probiotic groups, serum TNF-α levels decreased significantly when compared with the NASH group (45.60 ± 2.17 pg/mL in the single-strain group and 46.46 ± 3.04 pg/mL in the mixed-strain group, *p* < 0.001 for both comparisons). Serum TNF-α levels in both probiotic groups were comparable to those in the control group. 

As illustrated in Figure 3B, serum IL-6 levels were significantly higher in the NASH group than in the control group (262.71 ± 12.49 vs. 199.57 ± 23.69 pg/mL, respectively, *p* < 0.05). In both probiotic groups, serum IL-6 levels decreased significantly when compared with the NASH group (209.42 ± 17.55 pg/mL in the single-strain group and 214.57 ± 6.53 pg/mL in the mixed-strain group, *p* < 0.05 for both comparisons). Serum IL-6 levels in both probiotic groups were comparable to those in the control group. 

### 3.4. Effects of Single- and Mixed-Strain Probiotics on CD14 and TLR4 Expressions

As shown in Figure 4, CD14 positivity increased significantly in the NASH group when compared with the control group (60.00 ± 9.00 vs. 14 ± 3.00, respectively; *p* < 0.05). CD14 positivity significantly decreased in single- and mixed-stain groups when compared with the NASH group (18.00 ± 1.00 vs. 29.00 ± 5.00 vs. 60.00 ± 9.00, respectively; *p* < 0.05). There was no statistically significant difference in CD14 positivity between both treatment groups and the control group (18.00 ± 1.00 vs. 29.00 ± 5.00 vs. 14 ± 3.00, respectively).

As shown in Figure 5, TLR4 positivity increased significantly in the NASH group when compared with the control group (45.00 ± 4.00 vs. 5.00 ± 2.00, respectively; *p* < 0.05). TLR4 positivity significantly decreased in single- and mixed-strain groups when compared with the NASH group (16.00 ± 1.00 vs. 20.00 ± 2.00 vs. 45.00 ± 4.00, respectively; *p* < 0.05). However, TLR4 positivity remained significantly higher in both single- and mixed- strain groups than in the control group (16.00 ± 1.00 vs. 20.00 ± 2.00 vs. 5.00 ± 2.00, respectively; *p* < 0.05).

### 3.5. Effects of Single- and Mixed-Strain Probiotics on Gut Microbiota Change

Gut microbiota analyses showed that the alpha diversity (Shannon diversity index) was reduced in the NASH group when compared with the control group (shown in Figure 6A–D). After probiotic intervention in the single-stain group, the alpha diversity significantly increased compared with the NASH group. The alpha diversity also increased in the mixed-strain group, but did not reach statistical significance compared to the NASH group. The beta diversity significantly differed in the NASH group as compared with the control group (shown in Figure 6B, PERMANOVA; *p* < 0.001). After probiotic intervention in both single- and mixed-stain groups, the beta diversity slightly shifted toward the control group (Figure 6E and Appendix A).

The distribution patterns of the top 20 relative abundances of bacteria at the genus level are shown in Figure 7A. To identify specific microbial communities that differed significantly among the groups, LEfSe analysis was performed (based on a cutoff of LDA score > 3.0, *p*-value < 0.005, and FDR-adjusted *p*-value < 0.1). Figure 7B shows that the main microbial species that differed between the control and other groups were *Lactobacillus* spp. and *Ruminococcus* spp. The main microbial species that differed between the NASH and other groups were *Akkermansia* spp., *Roseburia* spp., and *Alistipes* spp. The main microbial species that differed between the single probiotics group and other groups were *Clostridium* spp., *Blautia* spp., *Lachnoclostridium* spp., and *Bifidobacterium* spp. The main microbial species that differed between the mixed probiotics group and other groups were *Allobaculum* spp., *Bacteroides* spp., *Parabacteroides* spp., and *Phascolarctobacterium* spp. These results indicated that these bacteria played a significant role in each group. Other significantly differentiated bacteria between groups are shown in Appendix A.

Additionally, we evaluated the associations between microbial compositions and other NASH parameters (shown in Figure 7D,E). We found that the decrease in *Ruminococcus bromii* was associated with increased hepatic free fatty acid levels, and the decrease in *Lactobacillus reuteri* was associated with increased serum IL-6 levels. Moreover, we found that the increase in *Alistipes* spp. was associated with increased hepatic free fatty acid levels, and the increase in *Clostridium celatum* was associated with higher NASH pathological scores.

Moreover, to identify potential microbial functions associated with probiotic supplementation, functional prediction analysis was performed using PICRUSt2. As shown in Figure 8, twenty-one pathways were found to be altered in the probiotics supplement group compared to the NASH group. Following probiotic supplementation, there was enrichment in 7 pathways, including streptomycin biosynthesis, alanine, aspartate and glutamate metabolism, glutathione metabolism, biosynthesis of vancomycin group antibiotics, galactose metabolism, zeatin biosynthesis, and fructose and mannose metabolism.

## 4. Discussion

Our innate immune system recognizes preserved bacterial cell-wall components, such as LPS and lipoproteins, alternatively called pathogen-associated molecular patterns (PAMPs), by pattern recognition receptors such as CD14 and TLR4 [25]. Gut barrier dysfunction in NAFLD subsequently leads to the leakage of bacteria or bacterial products into the circulation. These PAMPs then activate Kupffer cells in the liver through CD14 and TLR4 and increase the production of TNF-α, IL-6, and C-C Motif Chemokine Ligand 2 (CCL2), which promote liver inflammation and fibrogenesis [26,27]. A previous animal study showed that the M1 to M2 (M1/M2) macrophage ratio had a positive correlation with plasma LPS binding protein levels and TLR4 and CD14 mRNA levels, supporting the role of endotoxemia in the induction of liver inflammation [28]. Moreover, the gut microbiome may play a role in the development of obesity-induced NAFLD, as a previous study has shown that genetically obese mice have increased intestinal permeability, higher circulating LPS and IL-6 levels, increased membrane CD14 expression on hepatic stellate cells (HSCs), and stronger inflammatory responses of HSCs to LPS than lean mice [9]. Similarly to prior studies, our results also demonstrate the increased number of TLR4- and CD14-positive cells and the increased serum TNF-α and IL-6 levels in the NASH group, supporting the potential role of PAMPs in the development of NASH.

Probiotics are live microorganisms that, when administered in appropriate amounts, confer beneficial effects to the host and contribute to the regulation of immune responses. *L. plantarum* has been shown to have direct anti-microbial effects against pathogenic bacteria, such as *Pseudomonas aeruginosa* and *Klebsiella pneumoniae* [29]. Moreover, several strains of *L. plantarum* have been found to have cholesterol-lowering capacity through cholesterol assimilation [30]. Moreover, in an animal model of alcoholic liver disease, *L. plantarum* could reduce serum TNF-α, IL-6, LPS, and hepatic malondialdehyde levels as compared with those in alcohol-fed rats alone [16]. Its antimicrobial effects against pathogenic gut bacteria, as well as its cholesterol-lowering, anti-inflammatory, and antioxidant properties observed in other studies, might explain our results, which showed that the *L. plantarum* supplement could improve NASH histopathology, reduce the number of TLR4- and CD14-positive cells in the liver, and lower serum TNF-α and IL-6 levels. We also observed similar findings that *L. plantarum* supplementation could improve gut microbial diversity and promote the growth of beneficial bacteria in rats with NASH. In accordance with our results, Cao and colleagues also demonstrated that *L. plantarum* supplementation alleviated HFD-induced hepatic steatosis and inflammation through the increased expression of hepatic fatty acid oxidation genes, the reduction in insulin resistance and ileal inflammation, and the improvement of gut dysbiosis [31].

In this study, we also found that the combination of *L. rhamnosus* L34 and *L. paracasei* B13 supplement (mixed-strain probiotics) reduced hepatic fat accumulation and NASH activity scores, normalized serum TNF-α levels, and lowered hepatic CD14 and TLR4 positivity. Moreover, our results showed that mixed-strain probiotics improved alpha diversity and enriched *Lactobacillus* spp. in the gut. The improvement of gut dysbiosis likely explains the reduction in Kupffer cell activation, as evidenced by the lower levels of hepatic CD14 and TLR4 expression and, thus, reduced liver inflammation. Although there were no studies that evaluated the effects of combined *L. rhamnosus* L34 and *L. paracasei* B13 supplementation on NASH, studies that used either *L. rhamnosus* or *L. paracasei* alone or in combination with other probiotics have shown positive results. Kim and colleagues demonstrated that *L. rhamnosus* GG supplementation reduced hepatic steatosis through the suppression of lipogenic gene expression, such as PPAR-gamma and Sterol regulatory element-binding protein 1c (SREBP1c). The authors also found that *L. rhamnosus* GG supplementation decreased mRNA levels of macrophage markers (i.e., F4/80 and CD11b) and IL-6 [32]. Similarly, Okubo and colleagues showed that *L. casei* supplement improved all components of NASH activity scores and liver fibrosis in a mouse model of methionine-choline-deficient (MCD) diet-induced NASH. They also found that *L. casei* supplementation suppressed hepatic SREBP1c gene expression, reduced serum LPS and hepatic TNF-α mRNA levels, and improved colonic inflammation [33]. Likewise, Wagnerberger and colleagues found that *L. casei* supplementation improved hepatic steatosis and normalized hepatic TLR4 expression in fructose-fed mice [34]. We hypothesized that mixed-strain probiotics improved NASH histopathology through the modulation of gut microbiota and the reduction in bacterial translocation, thus decreasing Kupffer cell activation and inflammatory cytokine production. Moreover, previous studies have suggested that mixed-strain probiotics might improve hepatic steatosis through the suppression of lipogenic gene expression as well.

A previous animal study showed that NASH was associated with the reduction in overall bacterial density and alpha diversity [35]. Similarly, our study demonstrated that the alpha diversity was reduced in the NASH group as compared with the control group and improved after both types of probiotic supplement. Moreover, HFD has been shown to affect gut microbial composition. For example, previous reports showed that HFD promoted an enrichment in some species of *Bacteroides* and *Alistipes,* but a reduction in *Lactobacillus* spp. [36,37,38]. Similarly, our study showed a significant increase in the relative abundance of *Bacteroides* spp. and *Alistipes* spp., but a significant decrease in the relative abundance of *Lactobacillus* in the NASH group as compared with the control group. In this study, we also found an association between the increase in *Alistipes* spp. and higher hepatic free fatty acid levels, suggesting the potential role of *Alistipes* spp. in fatty liver disease. In contrast to our findings, a significant reduction in *Alistipes* spp. was observed in patients with NASH, particularly those with significant fibrosis [39]. Moreover, we observed significant enrichment of *Clostridium* spp. in the NASH group and a correlation between an increase in *Clostridium celatum* and higher NASH pathological scores. A previous study similarly demonstrated the enrichment of *Clostridium* spp. in mice fed with a high-cholesterol diet. The authors also found a positive correlation between *Clostridium celatum* and hepatic cholesterol levels [40]. Interestingly, we found a significant increase in the relative abundance of *Akkermansia* spp. in the NASH group as compared with the control group. A similar finding was reported by another study using a mouse model of a high-fat and high-sucrose diet [41]. That study reported a negative correlation between body fat percentage growth and the abundance of *Akkermansia* spp., but a positive correlation with the relative abundance of *Lactobacillus* spp. [42,43]. We hypothesized that body weight loss in the NASH group might explain the increase in the relative abundance of *Akkermansia* spp. in that group as compared with the control group.

At the genus level, the relative abundances of *Lactobacillus* spp. and *Allobaculum* spp. were significantly elevated after treatment with single and mixed strains of probiotics compared with the NASH group. A previous study has shown a similar result, evidencing that *Firmicutes* and the butyrate level were significantly elevated after the intervention with *Lactobacillus* spp. [44]. Lactate from *Lactobacillus* spp. could increase butyrate-producing bacterial strains [45], therefore promoting butyrate production in feces and the growth of other beneficial bacteria. Moreover, most species in the genus *Allobaculum* are short-chain fatty acid producers and could potentially protect against obesity and insulin resistance [46]. A previous study has shown that the ratio of *Bacteroidetes*/*Firmicutes* in the HFD group increased when compared to the normal diet group and decreased after the administration of *Lactobacillus* symbiotic [37]. Another animal study demonstrated that *L. plantarum* Q16 improved liver histology and metabolic parameters, reduced inflammatory markers, and suppressed de novo lipogenesis gene expression in mice fed with an HFD. The authors also showed the increased relative abundance of *Lactobacillus* spp. and decreased relative abundance of *Akkermansia* spp. in *L. plantarum* Q16-treated mice [47]. Similar findings were also observed in our study. Wang and colleagues showed that *L. rhamnosus* treatment in NAFLD mice improved hepatic steatosis, increased fecal short-chain fatty acid concentrations, decreased the relative abundance of *Clostridium* spp., and increased the abundance of *Bifidobacterium* spp. in the gut [48]. We similarly found an increase in the relative abundance of *Bifidobacterium* spp. in the mixed-strain group compared to the NASH group. Pan and colleagues demonstrated that postbiotics prepared using *Lactobacillus paracasei* improved hepatic steatosis and lobular inflammation in HFD-fed mice. However, in contrast to our study, the authors found that postbiotics increased the *Bacteroidetes*/*Firmicutes* ratio and the relative abundance of *Akkermansia* spp. as compared to the HFD group. Moreover, the functional prediction analysis of our study showed that probiotic supplement was associated with the glutathione pathway, which is a major pathway for the synthesis of natural antioxidants. We hypothesized that *Lactobacillus* supplementation, either with single or mixed strains, would promote the growth of other beneficial bacteria, leading to the improvement of gut dysbiosis, the reduction in LPS exposure to the liver, and the amelioration of inflammation and oxidative stress in the liver, eventually leading to the alleviation of NASH.

## 5. Conclusions

This study demonstrated that single and mixed strains of probiotics are effective in reducing the severity of NASH, likely through the modulation of gut microbiota, which led to the alleviation of hepatic lipid accumulation and inflammation. To the best of our knowledge, this is the first study to demonstrate the treatment effects of the combination of *L. rhamnosus* L34 and *L. paracasei* B13 supplements in an animal model of NASH. Clinical studies are needed to confirm the therapeutic effects of this combination in humans.

## Figures and Tables

**Figure 1 biomedicines-12-01847-f001:**
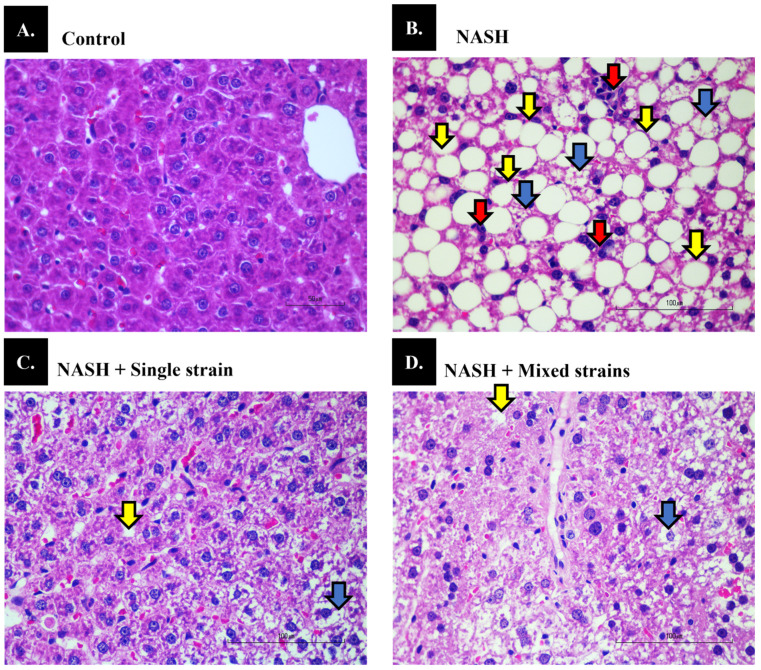
Effects of single and mixed strains of probiotics on liver histopathological changes. (**A**) Control group, (**B**) NASH group, (**C**) NASH + single-strain group and (**D**) NASH+mixed-strain group. Hematoxylin and eosin staining (400×, scale bar 100 µm and *n* = 7 per group). Yellow arrows indicate steatosis, red arrows indicate lobular inflammation, and blue arrows indicate hepatocyte ballooning.

**Figure 2 biomedicines-12-01847-f002:**
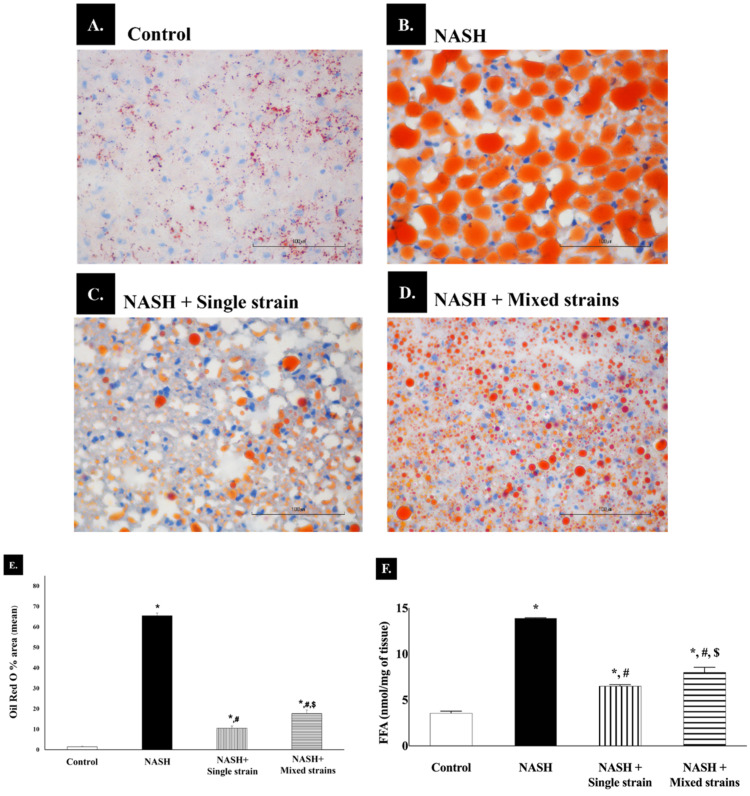
Effects of single and mixed strains of probiotics on hepatic lipid accumulation. (**A**–**D**): Histological images of Oil Red O staining in each group (400×, scale bar 100 µm and *n* = 7 per group). (**E**) Bar graph representing the percentage of Oil Red O staining area in each group. (**F**) Bar graph representing the hepatic free fatty acid levels in each group. Data are expressed as mean ± SEM. * *p* < 0.05 compared with the control group; ^#^ *p* < 0.05 compared with the NASH group; ^$^ *p* < 0.05 compared with the single-strain group.

**Figure 3 biomedicines-12-01847-f003:**
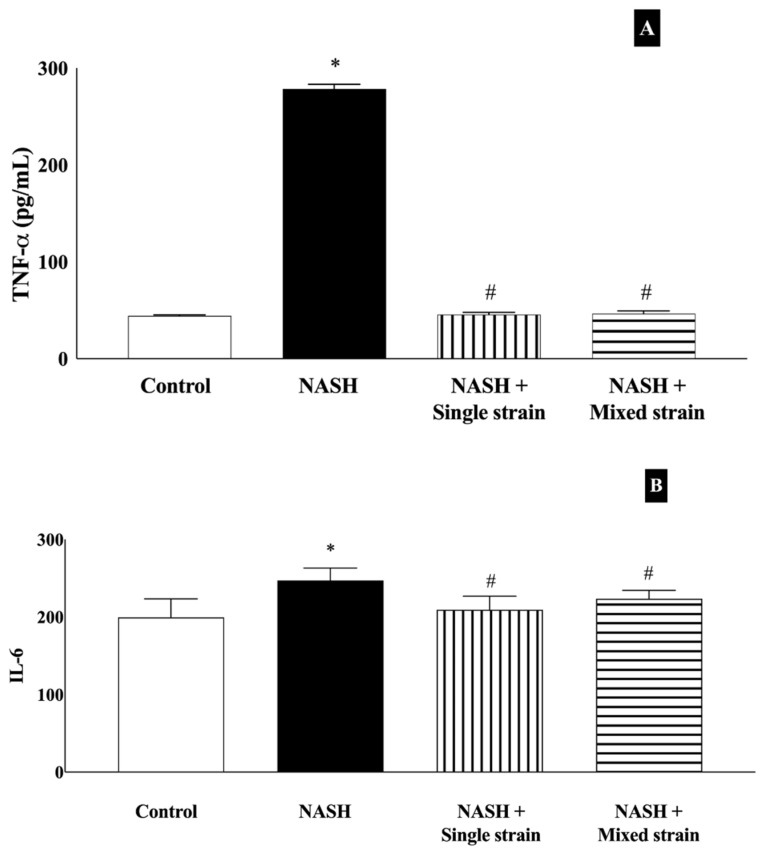
Effects of single and mixed strains of probiotics on serum tumor necrosis factor-a (**A**) and interleukin-6 (**B**). Data are expressed as mean ± SEM. * *p* < 0.05 compared with the control group; ^#^ *p* < 0.05 compared with the NASH group.

**Figure 4 biomedicines-12-01847-f004:**
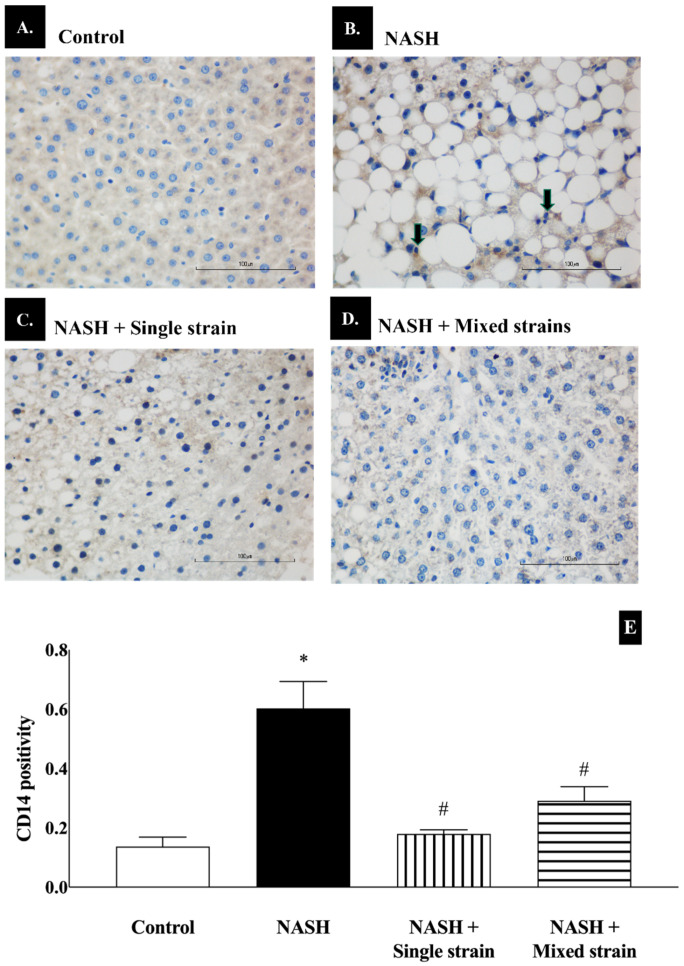
Effects of single and mixed strains of probiotics on CD14 expression. (**A**–**D**) Representative images of CD14 expression in each group (400×, scale bar 100 µm and *n* = 7 per group). (**E**): Bar graph representing the quantitative measurement of CD14 positivity in each group. Black arrows indicate Kupffer cells with brown-stained cytoplasm. Data are expressed as mean ± SEM. * *p* < 0.05 compared with the control group; ^#^ *p* < 0.05 compared with the NASH group.

**Figure 5 biomedicines-12-01847-f005:**
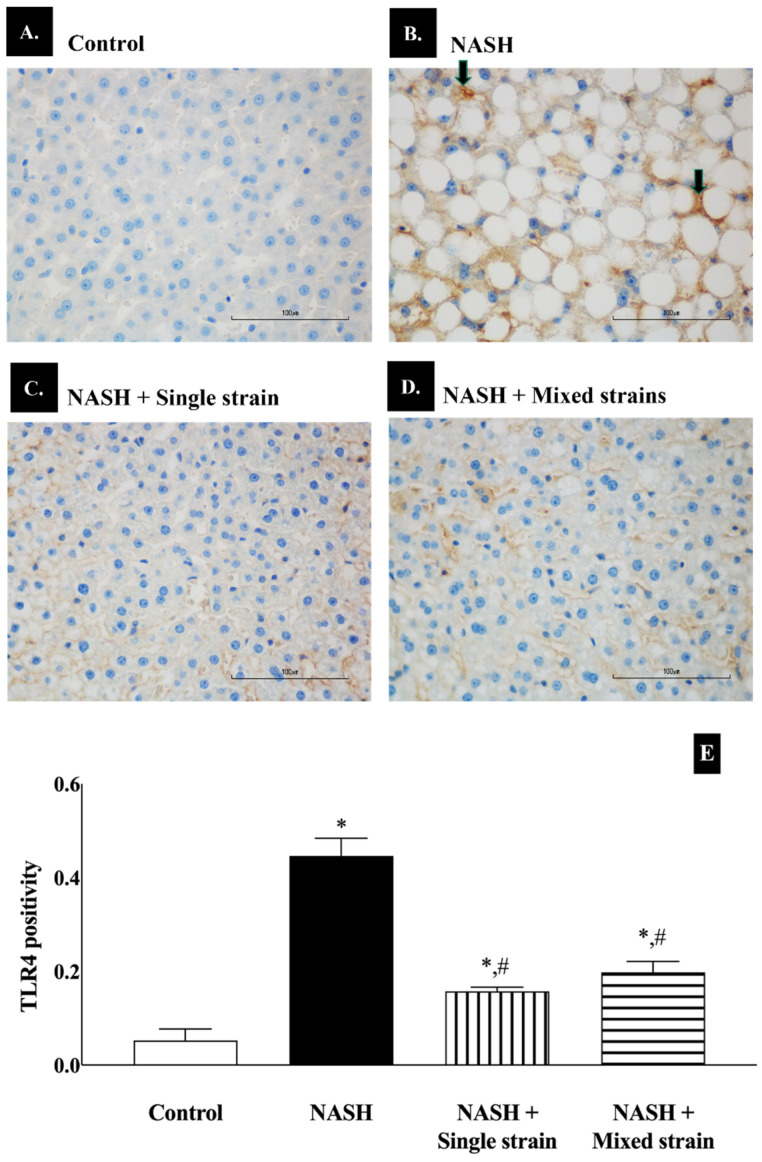
Effects of single and mixed strains of probiotics on toll-like receptor 4 expression. (**A**–**D**) Representative images of TLR4 expression in each group (400×, scale bar 100 µm and *n* = 7 per group). (**E**) Bar graph representing the quantitative measurement of TLR4 positivity in each group. Black arrows indicate hepatocytes with brown-stained cytoplasm. Data are expressed as mean ± SEM. * *p* < 0.05 compared with the control group; ^#^
*p* < 0.05 compared with the NASH group.

**Figure 6 biomedicines-12-01847-f006:**
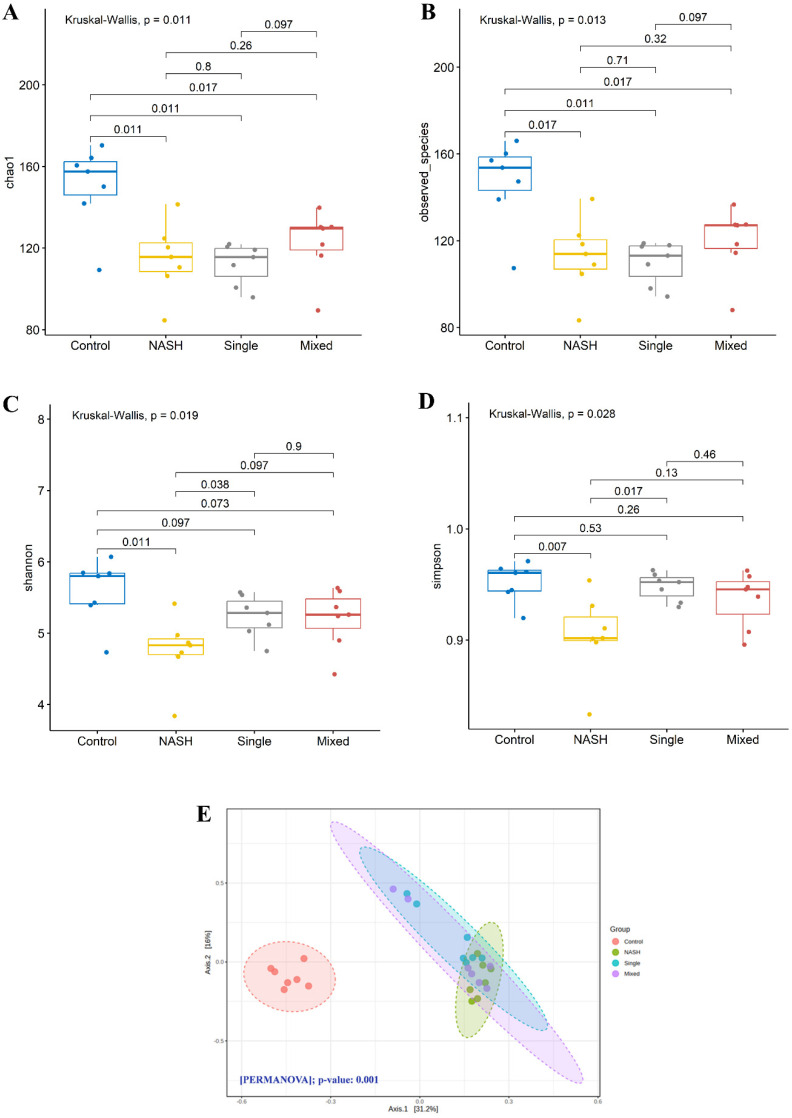
Effects of single- and mixed-strain probiotics on alpha and beta diversity of gut microbiota (**A**) Chao1 index; (**B**) observed species index; (**C**) Shannon index; (**D**) Simpson index; (**E**) principal coordinates analysis (PCoA, based on Bray–Curtis distance).

**Figure 7 biomedicines-12-01847-f007:**
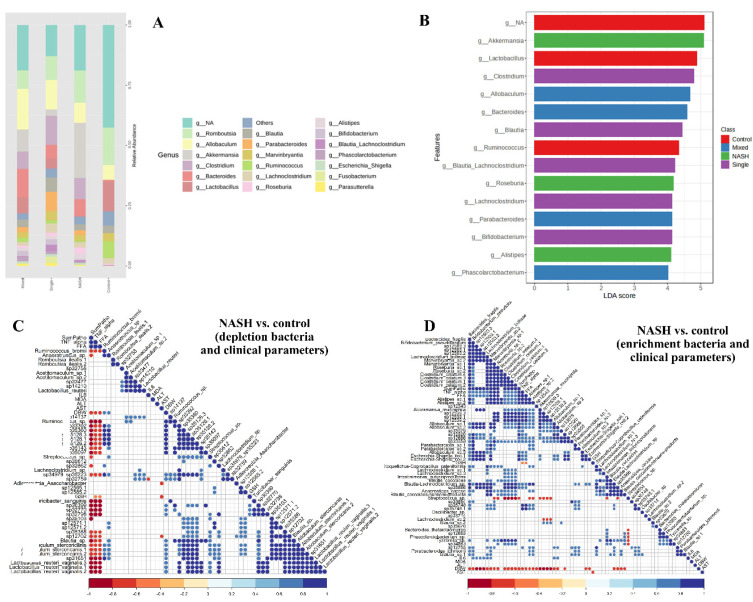
Effects of single and mixed strains of probiotics on gut microbiota composition. (**A**) Top 20 relative abundances of gut microbial composition at the genus level in each group; (**B**) LEfSe analysis among groups (cutoff of LDA score > 3.0, *p*-value < 0.005 and FDR-adjusted *p*-value < 0.1); (**C**) the associations between depleted microbial compositions and clinical factors; (**D**) the enrichment of bacteria in NASH vs. control and the correlation with clinical parameters (Spearman’s correlation with *p* < 0.005)”. The blue dot indicates a positive correlation between the clinical parameter on the diagonal axis and the microorganism on the x-axis. The red dot indicates a negative correlation between the clinical parameter on the diagonal axis and the microorganism on the x-axis. The higher the intensity of the dot color, the higher the r-value, as indicated by the color chart below the graph.

**Figure 8 biomedicines-12-01847-f008:**
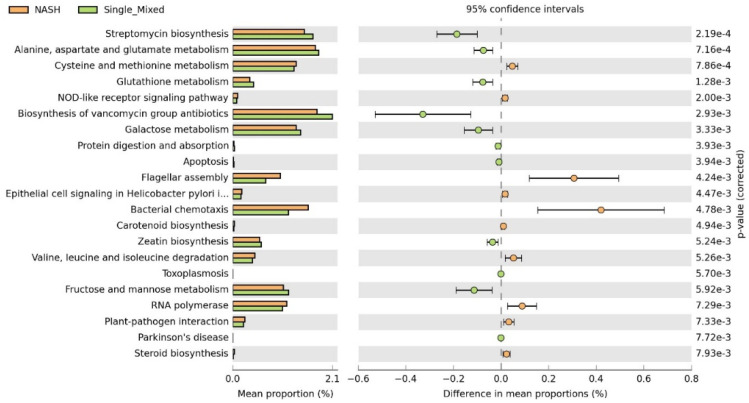
Functional prediction of gut microbial community by PICRUSt2 analysis. Significantly enriched KEGG pathways were found between the NASH and probiotics groups (*p*-value < 0.001).

**Table 1 biomedicines-12-01847-t001:** The histological scores in each group.

Group	*n*	Steatosis				Inflammation				Hepatocyte Ballooning			NASH Score
		0	1	2	3	0	1	2	3	0	1	2	
Control	7	7	-	-	-	6	1	-	-	6	1	-	0 ± 0
NASH	7	-	-	1	6	-	-	6	1	-	-	7	7 ± 0 *
NASH + Single Strain	7	1	2	3	1	3	4	-	-	-	-	7	5 ± 2 *^,#^
NASH + Mixed Strains	7	-	3	3	1	1	6	-	-	-	2	5	5 ± 2 *^,#^

Data are expressed as the number of rats in each histology grade. NAS scores are presented as median ± interquartile range. * *p*-value < 0.05 when compared with the control group; ^#^ *p*-value < 0.05 when compared with the NASH group. Steatosis grade: 0 = absence of fat, 1 = < 33% of hepatocytes with fat droplets, 2 = 33–66% of hepatocytes with fat droplets, 3 = > 66% of heap atocytes with fat droplets. Inflammation grade; 0 = no inflammation, 1 = < 2 foci per 20 × field, 2 = 2–4 foci per 20 × field, 3 = > 4 foci per 20 × field. Hepatocyte ballooning grade: 0 = no ballooning, 1 = few ballooning cells, 2 = many ballooning cells.

## Data Availability

The original contributions presented in the study are included in the article/Appendix A. Further inquiries can be directed to the corresponding author.

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
