# Peer review of "Single and Mixed Strains of Probiotics Reduced Hepatic Fat Accumulation and Inflammation and Altered Gut Microbiome in a Nonalcoholic Steatohepatitis Rat Model"

_biomedicines, 2024, doi:10.3390/biomedicines12081847_

Round 1

Reviewer 1 Report

Comments and Suggestions for Authors

1.       All Pin the article need to be in italics, and p or P needs to be consistent;

2.      Why did the author only choose TNF-alpha for blood biochemical indicators?  It is recommended to supplement some other inflammatory cytokines IL-2, IL-4, IL-6, etc. since the authors mentioned IL-6 in the manuscript.

3.      It is recommended that the author merge Figure 2 into one picture. In addition, the author should re-create all the figures in the manuscript by graphics tools such as Graphics.  High-quality pictures often improve the quality of the article.

4.      The abbreviations in the Figure legends should be written in their full names to facilitate readers understanding

5.      There are also other α-diversity indicators such as chao 1, Richness, Simpson, etc., what are the results? 

6.      Figure 6E and D, the associations between microbial compositions and clinical factors. Are values in Figure 6C represent the mean value?  It is recommended that the author supplement the explanation clearly in the Figure legend.

7.      Line 325-326: the authors found” the relative abundance of Lactobacillus significantly decreased in the NASH group compared with the control group and increased in both treatment groups.” Where are the statistically significant results? please add the Linear discriminant analysis effect size (LEfSe) analysis to identify specific microbial communities that differed significantly among the groups.

8.      Line 336: based on “Lactobacillus reuteri was associated with increased serum IL-6 levels”, and Figure 6 D, the associations between microbial compositions and other NASH parameters,  I did not found the results of IL-6ALTAST and MDA in the resultsPlease add on. Please add on.  It is recommended to reanalyze the correlation analysis with reference to relevant literature, and elaborate in the results section which microorganisms are positively or negatively correlated with NASH indicators.

9.  I personally suggest that the author conducts an in-depth analysis of the results of 16s RNA and improves the results and discussion sections based on the new analysis results.

Author Response

Reviewer 1

Comments and Suggestions for Authors

  1.      All “P”in the article need to be in italics, and p or P needs to be consistent;

Ans: We changed all “p” to “p” as suggested.

  1.     Why did the author only choose TNF-alpha for blood biochemical indicators?  It is recommended to supplement some other inflammatory cytokines IL-2, IL-4, IL-6, etc. since the authors mentioned IL-6 in the manuscript.

Ans: We have added the results of serum IL-6 levels in the revised manuscript and highlighted in yellow.

  1.     It is recommended that the author merge Figure 2 into one picture. In addition, the author should re-create all the figures in the manuscript by graphics tools such as Graphics.  High-quality pictures often improve the quality of the article.

Ans: We have merged Figure 2 into one picture as suggested. Our bar graphs were created using GraphPad Prism.

  1.     The abbreviations in the Figure legends should be written in their full names to facilitate readers understanding

Ans: We have changed the parameters’ names from abbreviations to their full names in the Figure legends as suggested.

  1.     There are also other α-diversity indicators such as chao 1, Richness, Simpson, etc., what are the results? 

Ans: We appreciate your comment. The Chao1 and Observed species, which represent community richness and Shannon and Simpson represent richness and evenness have been added in this revised manuscript as Figure 6. The alpha diversity and beta diversity have been combined in the same figure.

  1.     Figure 6E and D, the associations between microbial compositions and clinical factors. Are values in Figure 6C represent the mean value?  It is recommended that the author supplement the explanation clearly in the Figure legend.

Ans: Thank you for your suggestion. In this revised manuscript, Figure 6 has been changed to Figure 7 and new Figure Legend has been added as follows.

Figure 7. Effects of single and mixed strains of probiotics on gut microbiota composition

A, Top 20 relative abundances of gut microbial composition at the genus level in each group; B, LEfSe analysis among groups (cutoff of LDA score >3.0, p-value <0.005 and FDR-adjusted p-value <0.1); C, associations between depleted microbial compositions and clinical factors; D, the enrichment of bacteria in NASH vs. control and correlation with clinical parameters (Spearman’s correlation with p <0.005)”. The blue dot indicates a positive correlation between the clinical parameter on the diagonal axis and the microorganism on the x-axis. The red dot indicates a negative correlation between the clinical parameter on the diagonal axis and the microorganism on the x-axis. The higher the intensity of the dot color, the higher the r-value, as indicated by the color chart below the graph.

  1.     Line 325-326: the authors found “the relative abundance of Lactobacillus significantly decreased in the NASH group compared with the control group and increased in both treatment groups.” Where are the statistically significant results? please add the Linear discriminant analysis effect size (LEfSe) analysis to identify specific microbial communities that differed significantly among the groups.

Ans: We appreciate the comments. To determine differentially abundant bacteria between two groups, DESeq2 analysis was performed and shown in Supplementary Figure 4, 5 and 6. However, in this revised manuscript, the LEfSe analysis among groups at genus level was performed as reviewer’s suggestion. The LEfSe analysis has been added in the methodology section and highlighted in yellow. We use LDA score >3.0, p-value <0.005 and FDR-adjusted p-value <0.1 for this analysis. The result has been added in the manuscript as follows.

“To identify specific microbial communities that differed significantly among the groups, LEfSe analysis was performed (based on cutoff of LDA score >3.0, p-value <0.005 and FDR-adjusted p-value <0.1). Figure 7B showed that the main microbial species that differed between the control and other groups were Lactobacillus spp. and Ruminococcus spp. The main microbial species that differed between the NASH and other groups were Akkermansia spp., Roseburia spp., and Alistipes spp. The main microbial species that differed between the single probiotics group and other groups were Clostridium spp., Blautia spp., Lachnoclostridium spp., and Bifidobacterium spp. The main microbial species that differed between the mixed probiotics group and other groups were Allobaculum spp., Bacteroides spp., Parabacteroides spp., and Phascolarctobacterium spp. These results indicated that these bacteria played a significant role in each group.”

  1.     Line 336: based on “Lactobacillus reuteri was associated with increased serum IL-6 levels”, and Figure 6 D, the associations between microbial compositions and other NASH parameters,  I did not found the results of IL-6、ALT、AST and MDA in the results,Please add on. It is recommended to reanalyze the correlation analysis with reference to relevant literature, and elaborate in the results section which microorganisms are positively or negatively correlated with NASH indicators.

Ans: The results of ALT, AST and MDA were previously published in the online proceeding book of the local scientific meeting. To avoid duplication of data, we decided not to include those results in the current manuscript. However, we have presented the data here for your review.

Control

NASH

NASH+Single strain

NASH+Mixed strain

p-value

AST

125.3±23.9

193.7±25.0

112.7±21.2

94.1±30.9

ns*, 0.03**, 0.01#

ALT

35.2±6.2

33.0±5.4

28.7±8.3

24.5±8.9

ns*,**,#

MDA

1.1±0.4

3.3±1.3

0.8±0.4

1.0±0.2

0.04*, 0.02**, 0.03#

*p-value comparing between Control and NASH groups; **p-value comparing between NASH and NASH+Single strain groups #p-value comparing between NASH and NASH+Mixed strain groups. ns= non-significant. Data are presented as mean±SEM.

The correlation between microorganisms and NASH indicators was presented in The Results section, page 22, line 413-419.

  1. I personally suggest that the author conducts an in-depth analysis of the results of 16s RNA and improves the results and discussion sections based on the new analysis results.

Ans: More details of gut microbiota analysis has been added in this revised manuscript. The alpha-diversity indies (Chao1, Observed species and Simpson), LEfSe analysis and Picrust2 analysis have been added into this version.

Moreover, to identify potential microbial functions associated with probiotic supplementation, functional prediction analysis was performed using PICRUSt2. As shown in Figure 8, 21 pathways were found to be altered in the probiotics supplement group compared to the NASH group. Following probiotic supplementation, there was enrichment in 7 pathways, including streptomycin biosynthesis, alanine, aspartate and glutamate metabolism, glutathione metabolism, biosynthesis of vancomycin group antibiotics, galactose metabolism, zeatin biosynthesis, and fructose and mannose metabolism.

Reviewer 1

Comments and Suggestions for Authors

  1.      All “P”in the article need to be in italics, and p or P needs to be consistent;

Ans: We changed all “p” to “p” as suggested.

  1.     Why did the author only choose TNF-alpha for blood biochemical indicators?  It is recommended to supplement some other inflammatory cytokines IL-2, IL-4, IL-6, etc. since the authors mentioned IL-6 in the manuscript.

Ans: We have added the results of serum IL-6 levels in the revised manuscript and highlighted in yellow.

  1.     It is recommended that the author merge Figure 2 into one picture. In addition, the author should re-create all the figures in the manuscript by graphics tools such as Graphics.  High-quality pictures often improve the quality of the article.

Ans: We have merged Figure 2 into one picture as suggested. Our bar graphs were created using GraphPad Prism.

  1.     The abbreviations in the Figure legends should be written in their full names to facilitate readers understanding

Ans: We have changed the parameters’ names from abbreviations to their full names in the Figure legends as suggested.

  1.     There are also other α-diversity indicators such as chao 1, Richness, Simpson, etc., what are the results? 

Ans: We appreciate your comment. The Chao1 and Observed species, which represent community richness and Shannon and Simpson represent richness and evenness have been added in this revised manuscript as Figure 6. The alpha diversity and beta diversity have been combined in the same figure.

  1.     Figure 6E and D, the associations between microbial compositions and clinical factors. Are values in Figure 6C represent the mean value?  It is recommended that the author supplement the explanation clearly in the Figure legend.

Ans: Thank you for your suggestion. In this revised manuscript, Figure 6 has been changed to Figure 7 and new Figure Legend has been added as follows.

Figure 7. Effects of single and mixed strains of probiotics on gut microbiota composition

A, Top 20 relative abundances of gut microbial composition at the genus level in each group; B, LEfSe analysis among groups (cutoff of LDA score >3.0, p-value <0.005 and FDR-adjusted p-value <0.1); C, associations between depleted microbial compositions and clinical factors; D, the enrichment of bacteria in NASH vs. control and correlation with clinical parameters (Spearman’s correlation with p <0.005)”. The blue dot indicates a positive correlation between the clinical parameter on the diagonal axis and the microorganism on the x-axis. The red dot indicates a negative correlation between the clinical parameter on the diagonal axis and the microorganism on the x-axis. The higher the intensity of the dot color, the higher the r-value, as indicated by the color chart below the graph.

  1.     Line 325-326: the authors found “the relative abundance of Lactobacillus significantly decreased in the NASH group compared with the control group and increased in both treatment groups.” Where are the statistically significant results? please add the Linear discriminant analysis effect size (LEfSe) analysis to identify specific microbial communities that differed significantly among the groups.

Ans: We appreciate the comments. To determine differentially abundant bacteria between two groups, DESeq2 analysis was performed and shown in Supplementary Figure 4, 5 and 6. However, in this revised manuscript, the LEfSe analysis among groups at genus level was performed as reviewer’s suggestion. The LEfSe analysis has been added in the methodology section and highlighted in yellow. We use LDA score >3.0, p-value <0.005 and FDR-adjusted p-value <0.1 for this analysis. The result has been added in the manuscript as follows.

“To identify specific microbial communities that differed significantly among the groups, LEfSe analysis was performed (based on cutoff of LDA score >3.0, p-value <0.005 and FDR-adjusted p-value <0.1). Figure 7B showed that the main microbial species that differed between the control and other groups were Lactobacillus spp. and Ruminococcus spp. The main microbial species that differed between the NASH and other groups were Akkermansia spp., Roseburia spp., and Alistipes spp. The main microbial species that differed between the single probiotics group and other groups were Clostridium spp., Blautia spp., Lachnoclostridium spp., and Bifidobacterium spp. The main microbial species that differed between the mixed probiotics group and other groups were Allobaculum spp., Bacteroides spp., Parabacteroides spp., and Phascolarctobacterium spp. These results indicated that these bacteria played a significant role in each group.”

  1.     Line 336: based on “Lactobacillus reuteri was associated with increased serum IL-6 levels”, and Figure 6 D, the associations between microbial compositions and other NASH parameters,  I did not found the results of IL-6、ALT、AST and MDA in the results,Please add on. It is recommended to reanalyze the correlation analysis with reference to relevant literature, and elaborate in the results section which microorganisms are positively or negatively correlated with NASH indicators.

Ans: The results of ALT, AST and MDA were previously published in the online proceeding book of the local scientific meeting. To avoid duplication of data, we decided not to include those results in the current manuscript. However, we have presented the data here for your review.

Control

NASH

NASH+Single strain

NASH+Mixed strain

p-value

AST

125.3±23.9

193.7±25.0

112.7±21.2

94.1±30.9

ns*, 0.03**, 0.01#

ALT

35.2±6.2

33.0±5.4

28.7±8.3

24.5±8.9

ns*,**,#

MDA

1.1±0.4

3.3±1.3

0.8±0.4

1.0±0.2

0.04*, 0.02**, 0.03#

*p-value comparing between Control and NASH groups; **p-value comparing between NASH and NASH+Single strain groups #p-value comparing between NASH and NASH+Mixed strain groups. ns= non-significant. Data are presented as mean±SEM.

The correlation between microorganisms and NASH indicators was presented in The Results section, page 22, line 413-419.

  1. I personally suggest that the author conducts an in-depth analysis of the results of 16s RNA and improves the results and discussion sections based on the new analysis results.

Ans: More details of gut microbiota analysis has been added in this revised manuscript. The alpha-diversity indies (Chao1, Observed species and Simpson), LEfSe analysis and Picrust2 analysis have been added into this version.

Moreover, to identify potential microbial functions associated with probiotic supplementation, functional prediction analysis was performed using PICRUSt2. As shown in Figure 8, 21 pathways were found to be altered in the probiotics supplement group compared to the NASH group. Following probiotic supplementation, there was enrichment in 7 pathways, including streptomycin biosynthesis, alanine, aspartate and glutamate metabolism, glutathione metabolism, biosynthesis of vancomycin group antibiotics, galactose metabolism, zeatin biosynthesis, and fructose and mannose metabolism.

Reviewer 2 Report

Comments and Suggestions for Authors

In this article, the authors have evaluated the effect of single and mixed Lactobacilli strains on the severity of the diet-induced rodent model of NASH. The paper is well-written and scientifically sound; however, I have some suggestions to enhance the quality of the work presented.

Fig 2: As shown in the representative pictures, the NASH group shows macrovesicular fat accumulation, whereas the probiotic-treated groups show microvesicular fat accumulation. If possible, please quantify macro vs. micro separately.

An increase in microvesicular droplets also could mean increased oxidation leading to oxidative stress and reactive oxygen species. Moreover, an increase in oxidative stress is also a characteristic of NAFLD/NASH. Therefore, an additional figure showing the reduction in oxidative stress markers by treatment with probiotic bacteria would strengthen the data.

Additionally, quantification of the triglyceride levels along with the FFA results (shown by the authors) would also strengthen the data.

Author Response

Reviewer 2

Comments and Suggestions for Authors

The authors studied the effect of single strain and mixed strain probiotics on the histological grade of NASH, expression of TLR4 and CD14 in the liver, as well as on serum TNF-α level. The conclusion of the present study is that the administration of L. plantarum B7 or of the combination of L. rhamnosus L34 and L. paracasei B13 reduces liver statosis and inflammation in HFHF-fed rats. The effect of these probiotics appears to be very robust according to this study. Although the role of gut dysbiosis in the pathogenesis of NAFLD is partly known, it seems unusual that only probiotic administration without dietary changes or physical activity could reduce steatosis and inflammation in the liver to such a high extent as the authors found in this study. It would be reasonable to determine some molecular mechanisms of NAFLD improvement such as the expression of SREBP-1c, oxidative stress, the expression of proinflammatory cytokines in the liver. Apart from this there are some other major shortcomings that should be addressed.

Ans: Thank you for your suggestion. We’ll study detailed molecular mechanisms in our future study.

Introduction

The aim of the study should be better defined. The last paragraph in the Introduction section should be re-written. First the state-of-the-art regarding these bacteria should be mentioned and then what is novel in this study should be better emphasized. The authors should then define in one sentence that the aim of the study was to determine the effects of L. plantarum B7, and the combination of L. rhamnosus L34 and L. paracasei B13 on NASH histopathology, liver CD14 and TLR4 expression, and serum TNF-α level in HFHF model in rats. In the last sentence of the Introduction section the authors state “With the combined effects of anti-inflammation and improved intestinal integrity, we hypothesized that the combination of L. rhamnosus L34 and L. paracasei B13 might strengthen gut barrier, reduce bacterial translocation and thus improving NASH histopathology.” This sentence has to be re-phrased since the authors did not determine gut barrier permeability and bacterial translocation in the present study.

Ans: We have maded some changes in the Introduction as suggested and highlighted in yellow.

Materials and Methods

Room temperature 25±1°C is quite high for keeping experimental animals.

Ans: This is the standard temperature used at our Animal Research Facility for rats.

Brunt's criteria for histological grading of NASH should be described in the Materials and Methods section. These criteria are explained below the table 1, but they should be defined in the Materials and Methods section and also the authors should explain how they standardized the counting of fat-laden hepatocytes (which part of the liver was analyzed, how many visual fields, which surface area).

Ans: We have added Brunt’s criteria in the Material and Methods section. For steatosis grading, the pathologist examined the entirety of four different histology slides in each group and estimated the percentage of hepatocytes with intracellular fat droplets. This is a standard steatosis grading in clinical settings as well. We have also added this information in the Material and Methods and highlighted in yellow.

Why did the authors determine total FFA level in the liver? The steatosis is confirmed by Oil Red O staining so the determination of total FFAs does not contribute significantly to the specification of steatosis. On the other side, different FFAs have different roles in the pathogenesis of NAFLD, so it would be useful to determine FFA profile in the liver. Total FFA level is also not significant in the revealing of the probiotic role in the pathogenesis of NAFLD.  

Ans: We used a percent positive pixel as a quantification method for Oil Red O staining, which is a semi-quantitative method. Therefore, we decided to complement this result with a more quantitative method, which is hepatic FFA. Also, Oil Red O stains both cholesterol and triglycerides, while triglycerides are a main player in the development of NAFLD. Therefore, we believed that hepatic FFA level better represents hepatic triglyceride accumulation.

The determination of liver TNF-α expression would be more useful for the determination of the role of probiotics on inflammation in the liver than serum TNF-α.

Ans: Thank you so much for your suggestion. We’ll consider this in our future project.

Results

Histological scores should be expressed as medians with 25th and 75th percentiles. A non-parametric test should be used for the analysis of pathohistological scores. * and # should be explained in the Legend for Table 1. It appears that there is a difference in CD14 expression between NASH+Mixed strains and control group, please check the statistical analysis once again.

Ans: We have now presented NAS score as median±IQR and used Kruskal-Wallis test with post-hoc Dunn’s test for the statistical analysis. We’ve also explained the symbols * and # in the Figure legend. We have repeated the statistical analysis as suggested and we would like to confirm that the difference in CD14 expression between NASH+Mixed strains and Control groups was not statistically significant.

Figure 6 must be explained in more details.

Ans: Figure 6 has been changed to Figure 7, and more detailed explanation has been added.

Discussion

Discussion section should be re-written with more emphasis on the authors’ results and less on the results of previous studies.

Ans: We have made some changes in the Discussion section as suggested.

Minor comments

Abbeviation HFHF should be defined when mentioned for the first time.

Ans: We have defined the abbreviation HFHF as high fat and high fructose both in the abstract and in the first paragraph of the Introduction section.

Comments on the Quality of English Language

English is very good, the article is easily readable. Some minor corrections are required.

Reviewer 3 Report

Comments and Suggestions for Authors

The authors studied the effect of single strain and mixed strain probiotics on the histological grade of NASH, expression of TLR4 and CD14 in the liver, as well as on serum TNF-α level. The conclusion of the present study is that the administration of L. plantarum B7 or of the combination of L. rhamnosus L34 and L. paracasei B13 reduces liver statosis and inflammation in HFHF-fed rats. The effect of these probiotics appears to be very robust according to this study. Although the role of gut dysbiosis in the pathogenesis of NAFLD is partly known, it seems unusual that only probiotic administration without dietary changes or physical activity could reduce steatosis and inflammation in the liver to such a high extent as the authors found in this study. It would be reasonable to determine some molecular mechanisms of NAFLD improvement such as the expression of SREBP-1c, oxidative stress, the expression of proinflammatory cytokines in the liver. Apart from this there are some other major shortcomings that should be addressed.

Introduction

The aim of the study should be better defined. The last paragraph in the Introduction section should be re-written. First the state-of-the-art regarding these bacteria should be mentioned and then what is novel in this study should be better emphasized. The authors should then define in one sentence that the aim of the study was to determine the effects of L. plantarum B7, and the combination of L. rhamnosus L34 and L. paracasei B13 on NASH histopathology, liver CD14 and TLR4 expression, and serum TNF-α level in HFHF model in rats. In the last sentence of the Introduction section the authors state “With the combined effects of anti-inflammation and improved intestinal integrity, we hypothesized that the combination of L. rhamnosus L34 and L. paracasei B13 might strengthen gut barrier, reduce bacterial translocation and thus improving NASH histopathology.” This sentence has to be re-phrased since the authors did not determine gut barrier permeability and bacterial translocation in the present study.

Materials and Methods

Room temperature 25±1°C is quite high for keeping experimental animals.

Brunt's criteria for histological grading of NASH should be described in the Materials and Methods section. These criteria are explained below the table 1, but they should be defined in the Materials and Methods section and also the authors should explain how they standardized the counting of fat-laden hepatocytes (which part of the liver was analyzed, how many visual fields, which surface area).

Why did the authors determine total FFA level in the liver? The steatosis is confirmed by Oil Red O staining so the determination of total FFAs does not contribute significantly to the specification of steatosis. On the other side, different FFAs have different roles in the pathogenesis of NAFLD, so it would be useful to determine FFA profile in the liver. Total FFA level is also not significant in the revealing of the probiotic role in the pathogenesis of NAFLD.  

The determination of liver TNF-α expression would be more useful for the determination of the role of probiotics on inflammation in the liver than serum TNF-α.

Results

Histological scores should be expressed as medians with 25th and 75th percentiles.  A non-parametric test should be used for the analysis of pathohistological scores. * and # should be explained in the Legend for Table 1. It appears that there is a difference in CD14 expression between NASH+Mixed strains and control group, please check the statistical analysis once again.

Figure 6 must be explained in more details.

Discussion
Discussion section should be re-written with more emphasis on the authors’ results and less on the results of previous studies.

Minor comments

Abbeviation HFHF should be defined when mentioned for the first time.

Comments on the Quality of English Language

English is very good, the article is easily readable. Some minor corrections are required.

Author Response

Reviewer 3

Comments and Suggestions for Authors

Chayanupatkul et al. explored the potential protective activity of a single-strain (L. plantarum) and a mixed-strain (L. rhamnosus + L. paracasei) probiotic supplements in rats with non-alcoholic steatohepatitis (NASH) induced using a high-fat, high-fructose diet for 6 weeks. Semi-quantitative liver histology, immunohistochemistry, serum biochemistry, and fecal microbiome metagenomic data were used to demonstrate that both supplements were similarly beneficial in terms of ameliorating hepatic steatosis and inflammation, and improving gut microbiome composition and diversity.

 Comments:

  1. Please clarify why L. plantarum itself wasn’t included in the combination given to the rats in the mixed-strain group. From a study design standpoint, would it be possibly rational to include a group given L. plantarum + L. rhamnosus + L. paracasei in a future follow-up study?

Ans: We did not combine L. plantarum B7 with L. rhamnosus L34 and L. paracasei B13 because, according to our unpublished data, L. plantarum B7 suppressed the growth of other bacteria when used in combination. We have added this statement in the Methods section.

  1. It appears from Materials & Methods that the rats in the single-strain group were receiving a total of 1.8×109 CFU/mL per single dosing, while the rats in the mixed-strain group were receiving a total of 3.6×109 CFU/mL (double the bacteria) per single dosing. If that is correct, was it considered that the different group results could at least partly be explained by the difference in dosing regimens and not only supplement composition (i.e., two overlapping factors, which makes ANOVA inapplicable)? If, however, 1.8×109 CFU/mL was the total dose given to both groups, a correction to the text should be made;

Ans: Rats in the mixed-strain group received the total of 1 mL of the combination suspension, which included L. rhamnosus L34 (1.8×109 CFU/mL) + L. paracasei B13 (1.8×109 CFU/mL) by oral gavage once daily (with a total bacteria dosage of 1.8×109 CFU/mL). We have made a correction in the text as suggested.

  1. NASH scores should be presented as ‘median; confidence interval’ instead of ‘mean; SEM’. Accordingly, the Kruskal-Wallis test or Mood’s median-test may be more appropriate to use for this kind of data;

Ans: NASH scores are now presented as median±IQR and the Kruskal-Wallis test was used for the statistical analysis.

  1. When used in a multiple-group setting, extended Fisher’s exact test requires a correction. Please specify whether such a correction was applied, and if so, which one;

Ans: We realized that Fisher’s exact test might not be the most appropriate test for multiple comparisons of histological grades; therefore, we compared the median score of each component of histological changes of NASH between groups using Kruskal-Wallis test with post-hoc Dunn’s test. We have added median score of each components in the manuscript.

  1. Were continuous numerical data tested for distribution normality prior to applying ANOVA? Please specify, and if so, also state the test used;

Ans: The Shapiro–Wilk test was used to test the normality of the continuous variables. Only NAS scores were not normally distributed and we have changed the statistical test to the Kruskal-Wallis test.

  1. Has the diet used to induce NASH been described and validated previously? If so, please provide a reference;

Ans: Yes, we used the formula for the HFHF diet that was previously validated in a study by our group. We have added this statement along with a reference in the Methods section.

  1. Probiotics are already known to provide benefits to NAFLD/NASH patients in the clinical setting. The conclusions should instead summarize whether a novel insight has been obtained into the details of probiotic strain composition, possible inter-strain synergism, mechanisms of action etc.;

Ans: We have made changes to the Conclusions as suggested.

  1. Whenever a whole genus of bacteria is mentioned, multiple species should be indicated with an spp. denotion; “…the relative abundance of Lactobacillus [species]…” should be changed to “…the relative abundance of Lactobacillus spp.…”, etc. throughout the text;

Ans: We have added “spp.” throughout the manuscript as appropriate.

  1. Line 444, ‘steasis’ should be corrected to ‘steatosis’;

Ans: We have corrected the typos as suggested.

  1. The abbreviations list should be expanded in order to include all currently missing abbreviations, i.e. LPS, HFD, SREBP1c, PPAR, PAMP, FFA, etc.

Ans: We have expanded the abbreviation list as suggested.

Comments on the Quality of English Language

In my opinion, English is of fairly good quality. However, I do not consider myself a great expert in this matter. It seems to me that it makes sense to focus on the opinion of a native speaker.

Reviewer 4 Report

Comments and Suggestions for Authors

Chayanupatkul et al. explored the potential protective activity of a single-strain (L. plantarum) and a mixed-strain (L. rhamnosus + L. paracasei) probiotic supplements in rats with non-alcoholic steatohepatitis (NASH) induced using a high-fat, high-fructose diet for 6 weeks. Semi-quantitative liver histology, immunohistochemistry, serum biochemistry, and fecal microbiome metagenomic data were used to demonstrate that both supplements were similarly beneficial in terms of ameliorating hepatic steatosis and inflammation, and improving gut microbiome composition and diversity.

 Comments:

1.        Please clarify why L. plantarum itself wasn’t included in the combination given to the rats in the mixed-strain group. From a study design standpoint, would it be possibly rational to include a group given L. plantarum + L. rhamnosus + L. paracasei in a future follow-up study?

2.        It appears from Materials & Methods that the rats in the single-strain group were receiving a total of 1.8×109 CFU/mL per single dosing, while the rats in the mixed-strain group were receiving a total of 3.6×109 CFU/mL (double the bacteria) per single dosing. If that is correct, was it considered that the different group results could at least partly be explained by the difference in dosing regimens and not only supplement composition (i.e., two overlapping factors, which makes ANOVA inapplicable)? If, however, 1.8×109 CFU/mL was the total dose given to both groups, a correction to the text should be made;

3.        NASH scores should be presented as ‘median; confidence interval’ instead of ‘mean; SEM’. Accordingly, the Kruskal-Wallis test or Mood’s median-test may be more appropriate to use for this kind of data;

4.        When used in a multiple-group setting, extended Fisher’s exact test requires a correction. Please specify whether such a correction was applied, and if so, which one;

5.        Were continuous numerical data tested for distribution normality prior to applying ANOVA? Please specify, and if so, also state the test used;

6.        Has the diet used to induce NASH been described and validated previously? If so, please provide a reference;

7.        Probiotics are already known to provide benefits to NAFLD/NASH patients in the clinical setting. The conclusions should instead summarize whether a novel insight has been obtained into the details of probiotic strain composition, possible inter-strain synergism, mechanisms of action etc.;

8.        Whenever a whole genus of bacteria is mentioned, multiple species should be indicated with an spp. denotion; “…the relative abundance of Lactobacillus [species]…” should be changed to “…the relative abundance of Lactobacillus spp.…”, etc. throughout the text;

9.        Line 444, ‘steasis’ should be corrected to ‘steatosis’;

10.    The abbreviations list should be expanded in order to include all currently missing abbreviations, i.e. LPS, HFD, SREBP1c, PPAR, PAMP, FFA, etc.

Comments on the Quality of English Language

In my opinion, English is of fairly good quality. However, I do not consider myself a great expert in this matter. It seems to me that it makes sense to focus on the opinion of a native speaker.

Author Response

Reviewer 4

Comments and Suggestions for Authors

Chayanupatkul et al. explored the potential protective activity of a single-strain (L. plantarum) and a mixed-strain (L. rhamnosus + L. paracasei) probiotic supplements in rats with non-alcoholic steatohepatitis (NASH) induced using a high-fat, high-fructose diet for 6 weeks. Semi-quantitative liver histology, immunohistochemistry, serum biochemistry, and fecal microbiome metagenomic data were used to demonstrate that both supplements were similarly beneficial in terms of ameliorating hepatic steatosis and inflammation, and improving gut microbiome composition and diversity.

 Comments:

  1. Please clarify why L. plantarum itself wasn’t included in the combination given to the rats in the mixed-strain group. From a study design standpoint, would it be possibly rational to include a group given L. plantarum + L. rhamnosus + L. paracasei in a future follow-up study?

Ans: We did not combine L. plantarum B7 with L. rhamnosus L34 and L. paracasei B13 because, according to our unpublished data, L. plantarum B7 suppressed the growth of other bacteria when used in combination. We have added this statement in the Methods section.

  1. It appears from Materials & Methods that the rats in the single-strain group were receiving a total of 1.8×109 CFU/mL per single dosing, while the rats in the mixed-strain group were receiving a total of 3.6×109 CFU/mL (double the bacteria) per single dosing. If that is correct, was it considered that the different group results could at least partly be explained by the difference in dosing regimens and not only supplement composition (i.e., two overlapping factors, which makes ANOVA inapplicable)? If, however, 1.8×109 CFU/mL was the total dose given to both groups, a correction to the text should be made;

Ans: Rats in the mixed-strain group received the total of 1 mL of the combination suspension, which included L. rhamnosus L34 (1.8×109 CFU/mL) + L. paracasei B13 (1.8×109 CFU/mL) by oral gavage once daily (with a total bacteria dosage of 1.8×109 CFU/mL). We have made a correction in the text as suggested.

  1. NASH scores should be presented as ‘median; confidence interval’ instead of ‘mean; SEM’. Accordingly, the Kruskal-Wallis test or Mood’s median-test may be more appropriate to use for this kind of data;

Ans: NASH scores are now presented as median±IQR and the Kruskal-Wallis test was used for the statistical analysis.

  1. When used in a multiple-group setting, extended Fisher’s exact test requires a correction. Please specify whether such a correction was applied, and if so, which one;

Ans: We realized that Fisher’s exact test might not be the most appropriate test for multiple comparisons of histological grades; therefore, we compared the median score of each component of histological changes of NASH between groups using Kruskal-Wallis test with post-hoc Dunn’s test. We have added median score of each components in the manuscript.

  1. Were continuous numerical data tested for distribution normality prior to applying ANOVA? Please specify, and if so, also state the test used;

Ans: The Shapiro–Wilk test was used to test the normality of the continuous variables. Only NAS scores were not normally distributed and we have changed the statistical test to the Kruskal-Wallis test.

  1. Has the diet used to induce NASH been described and validated previously? If so, please provide a reference;

Ans: Yes, we used the formula for the HFHF diet that was previously validated in a study by our group. We have added this statement along with a reference in the Methods section.

  1. Probiotics are already known to provide benefits to NAFLD/NASH patients in the clinical setting. The conclusions should instead summarize whether a novel insight has been obtained into the details of probiotic strain composition, possible inter-strain synergism, mechanisms of action etc.;

Ans: We have made changes to the Conclusions as suggested.

  1. Whenever a whole genus of bacteria is mentioned, multiple species should be indicated with an spp. denotion; “…the relative abundance of Lactobacillus [species]…” should be changed to “…the relative abundance of Lactobacillus spp.…”, etc. throughout the text;

Ans: We have added “spp.” throughout the manuscript as appropriate.

  1. Line 444, ‘steasis’ should be corrected to ‘steatosis’;

Ans: We have corrected the typos as suggested.

  1. The abbreviations list should be expanded in order to include all currently missing abbreviations, i.e. LPS, HFD, SREBP1c, PPAR, PAMP, FFA, etc.

Ans: We have expanded the abbreviation list as suggested.

Comments on the Quality of English Language

In my opinion, English is of fairly good quality. However, I do not consider myself a great expert in this matter. It seems to me that it makes sense to focus on the opinion of a native speaker.

Round 2

Reviewer 1 Report

Comments and Suggestions for Authors

no

Author Response

Comments and Suggestions for Authors : no

Thank you for giving us the opportunity to respond to your constructive comments. We are very grateful for the positive and helpful suggestions and we feel that the quality of the manuscript has been significantly improved.